# Damage Adaptive Titanium Alloy by In-Situ Elastic Gradual Mechanism

**DOI:** 10.3390/ma13020406

**Published:** 2020-01-15

**Authors:** Siqian Zhang, Jing Liu, Haoyu Zhang, Jie Sun, Lijia Chen

**Affiliations:** 1School of Materials Science and Engineering, Shenyang University of Technology, Shenyang 110870, China; zwtsysmm@163.com (J.L.); zhanghaoyu@sut.edu.cn (H.Z.); chenlijia@sut.edu.cn (L.C.); 2State Key Laboratory of Rolling and Automation, Northeastern University, Shenyang 110004, China; sunjie@ral.neu.edu.cn

**Keywords:** elastic gradient, fatigue damage tolerance, crack deflection, adaptive mechanism

## Abstract

Natural materials are generally damage adaptive through their multilevel architectures, with the characteristics of compositional and mechanical gradients. This study demonstrated that the desired elastic gradient can be in-situ stress-induced in a titanium alloy, and that the alloy showed extreme fatigue-damage tolerance through the crack deflection and branch due to the formation of a three-dimensional elastically graded zone surrounding the crack tip. This looks like a perceptive and adaptive mechanism to retard the crack: the higher stress concentrated at the tip and the larger elastic gradient to be induced. The retardation is so strong that a gradient nano-grained layer with a thickness of less than 2 μm formed at the crack tip due to the highly localized and accumulated plasticity. Furthermore, the ultrafine-grained alloy with the nano-sized precipitation also exhibited good damage tolerance.

## 1. Introduction

Graded materials have attracted great attention because they exhibit significant damage resistance that cannot be realized in conventional homogeneous materials [1,2]. Progress in material synthesis makes it more flexible to design and fabricate surface coatings having the desired gradient in composition, microstructure, and mechanical properties. For example, surface mechanical grinding treatment to fabricate gradient nano-grained copper which can sustain tensile strain over 100% without cracking, and sintering treatment to fabricate an elastic gradient coating on brittle ceramics that can avoid surface crack nucleation during indentation loading [3]. Furthermore, composite materials with periodic hybrid structures have been fabricated to improve the toughness of brittle materials [4,5].

Compared with these synthetic materials, metal materials are substantially heterogenetic where elastic and mechanical properties vary widely at anatomic sites and have complex hierarchical microstructures at many dimensional scales [6,7,8]. These natural materials exhibit formidable resistance to impact and fatigue damages [9,10,11]. Although they are extremely difficult to replicate synthetically, their damage-adaptive character thanks to their intricate architectures creates a guide to pioneer smart materials, which can detect the potential damage and then simultaneously trigger a defensive mechanism to ease the damage.

An elastic gradient could be a valid mechanism. Investigations have shown that coatings with a gradual increase of elastic modulus retarded fatigue crack growth by deflecting the crack from its nominal growth plane [12,13]. Such damage tolerance, achieved by an elastic gradient coating with gradual chemical compositions, is crucial to solid materials. Modern defect-tolerant design approaches to fatigue are based on the premise that engineering structures are inherently flawed, but solids have weak resistance to defects where cracks initiate preferentially and propagate at low-level stress intensity factors, and their growth rates increase with the increase of crack length until catastrophic failure [14,15].

Fortunately, we took notice that an in-situ elastic gradient can be triggered by an external applied stress for smart-solid materials such as natural materials, layered ceramics, hexagonal close-packed metals, titanium alloys, and micro/nano-sized crystals [16,17,18,19,20]. The aim of this study was to demonstrate that the in-situ stress-induced elastic gradient mechanism leads to extreme fatigue damage tolerance using a titanium alloy, Ti-10V-2Fe-3Al-0.11O (weight percent), as a model material.

## 2. Materials and Methods

### 2.1. Materials Preparation

Sponge Ti, pure Fe, and V-Al master alloys were used to prepare the alloy ingot. Raw materials were melted twice by vacuum arc remelting to ensure chemical homogeneity. The β-transus temperature of the alloy, verified by metallographic method, was 1088 K. The ingot, 140 mm in diameter and 20 mm in length was hot forged at 1123 K to a round bar 55 mm in diameter and then hot-rolled at 1073 K to 15 mm in diameter. The purity of the alloy after hot-rolling was 99.8%. The chemical compositions obtained by QL-S3000D wet chemical and HADPGM-7800VRAE gas analyses are shown in Table 1, and the tolerances were ±0.001%.

Ingots of Ti-10V-2Fe-3Al and Ti-10V-2Fe-3Al-0.82O alloys, with diameters of 120 mm, were also melted (Chemical compositions see Table 1). The ingot of Ti-10V-2Fe-3Al alloy was hot-forged at 1273 K and then at 1123 K into a round bar 25 mm in diameter. The purity of the alloy after hot-rolling was 99.7%. An ingot of Ti-10V-2Fe-3Al-0.82O alloy was forged at 1123 K into a billet 55 mm in diameter, cold-swaged at an initial temperature of 573 K into a rod with a diameter of 25 mm (the warm-swaging) and then cold-rolled at an initial temperature of 673 K to a diameter of 15 mm (the warm-rolling). The purity of the alloy was 99.8%.

### 2.2. Uniaxial Cyclic Tensile Tests

Uniaxial cyclic tensile tests were conducted by an MTS landmark 370.10 servohydraulic test system (MTS Industrial Systems, MN, USA) in air at room temperature (~295 K) using tensile specimens (Figure 1) by a cyclic loading–unloading deformation at an initial strain rate of 1.3 × 10^−4^ s^−1^. To ensure measurement accuracy, a strain gauge was used to record the cyclic stress–strain curves. The accuracy was confirmed by the consistent results of the initial Young’s modulus, measured from the stress–strain curve and the dynamical Young’s modulus measured by a free resonant vibration method (JE-RT Young’s modulus measurement apparatus, Beijing, China).

### 2.3. Fatigue Crack Growth Experiments

Fatigue crack growth tests were performed in air at room temperature by an MTS landmark 370.10 servohydraulic test system using single edge-cracked bend specimens with dimensions of 70 mm× 12mm × 5.5 mm (Figure 2) and single edge-cracked tension specimens with dimensions 80mm × 11mm × 2 mm (Figure 3), which was taken from the center of each ingot. Before the fatigue test, a 1-mm incision was cut in the center of these specimens using an electrospark wire-electrode machine (Figure 2 and Figure 3). Three-point bending fatigue tests were conducted at stress ratios *R* of 0.1, 0.3, and 0.5 with a cyclic frequency of 10 Hz, while the tension–tension fatigue tests conducted at an *R* of 0.1.

The descending stress method was used to determine the stress level for the fatigue crack growth which satisfied the crack length of less than 0.1 mm in 10^6^ cycles. Then, the pre-crack stress and the fatigue stress tests were selected as higher, 20% and 10% respectively, than the above-determined stress. After the pre-crack length reached about 1 mm, the fatigue test started to measure fatigue crack growth using a COD (crack opening displancement) strain gauge. The fatigue crack growth rate (d*a*/d*N*) and the stress intensity factor (Δ*K*) were determined by the MTS system program based on the flexibility method. 

The stress–strain field in front of the crack tip was monitored continuously during the fatigue tests using a strain gauge of 2mm × 3 mm, which was adhered about 1 mm in front of the crack tip and/or the incision. The variations of the strains were recorded using a YD-28 strain analyzer (MTS Industrial Systems, MN, USA) while the corresponding load changes were recorded by the MTS testing system until the strain gauge was destroyed due to the propagated crack.

### 2.4. Microstructure Characterizations

Scanning electron microscopy (SEM, Hitachi Su8010, Tokyo, Japan), transmission electron microscopy (TEM, JEOL JEM-2100, Tokyo, Japan), and electron back scattering diffraction (EBSD, Zeiss Gemini 500, Oberkochen, Germany) were used to investigate the microstructure evolution of alloys during fatigue. SEM specimens were prepared by electropolishing in a solution of 10 mL n-butanol, 90 mL ethanol, 6 g AlCl_3_, and 28 g ZnCl_2_ at a temperature of ~245 K. The EBSD image, with an accuracy of 50 nm, was operated at voltage of 20 kV and a current of 6.0 nA. TEM specimens with a crack tip were prepared using precision dimpling followed by ion milling with 5 keV argon ions at an incident angle of 15 degrees without a cold finger attachment and operated at 200 kV.

## 3. Results

### 3.1. Mechanical Properties

The Ti-10V-2Fe-3Al-0.11O alloy exhibited an elastic deformation behavior with the maximum recoverable strain up to 3.3% (Figure 4A). Most importantly, the novel nonlinear elasticity was triggered gradually from the start of loading (Figure 4(A1)), resulting in gradual elastic softening such that the tangent modulus decreased with the increase of tensile strain. This is in sharp contrast with conventional solids, exhibiting linear elasticity in advance of plastic deformation and/or phase transformation.

The Ti-10V-2Fe-3Al-0.82O alloy passed through martensite transformations (MTs) as the strain reached ~0.5% (Figure 4B) exhibited more significant nonlinear elasticity (Figure 4(B1)) and elastic softening in the low-strain range than the Ti-10V-2Fe-3Al-0.11O alloy. The following analyses demonstrated that the Ti-10V-2Fe-3Al-0.11O alloy exhibited extreme fatigue damage tolerance because the stress gradient surrounding the crack tip induced an elastic gradient to resist crack growth.

### 3.2. Fatigue Crack Propagation Pattern

Three-point bending tests were applied to investigate fatigue crack propagation using the single edge-cracked bend specimens of Ti-10V-2Fe-3Al, Ti-10V-2Fe-3Al-0.82O, and Ti-10V-2Fe-3Al-0.11O. Ti-10V-2Fe-3Al exhibited the normal fatigue crack growth behavior of metallic materials and its path was almost along the nominal growth plane (Figure 5A), while its growth rate (d*a*/d*N*) increased sharply with increasing stress intensity factor (Δ*K*) at the near-threshold region, and then increased stably at the Paris region (Figure 5D). Interestingly, the crack path of Ti-10V-2Fe-3Al-0.82O deviated significantly from the nominal plane at the beginning, propagated a long distance, then changed its direction toward the nominal plane (Figure 5B). Additionally, its continual growth also followed such behavior, but the deviation was much smaller. The significant crack deflection led to much better damage resistance as compared with Ti-10V-2Fe-3Al (Figure 5D), for example, the fatigue crack growth threshold (Δ*K*_0_) increased from 4.2 to 13.2 MPa/m^1/2^. Since Ti-10V-2Fe-3Al-0.82O alloy exhibits nonlinear elasticity at low stress levels, the crack deflection can be explained by the in-situ elastic gradient at the crack tip. This is in agreement with the previous studies of the elastic gradient compositie material [21,22].

Unexpectedly, the Ti-10V-2Fe-3Al-0.11O alloy exhibited a much more complicated crack pattern (Figure 5C), which appeared to be a group of deflected and branched cracks. They were enlarged in turn to show the details (Figure 5(C1–3)). This phenomenon is similar to what is seen in damage-tolerant natural materials with multilevel architectures [23]. Compared with the Ti-10V-2Fe-3Al-0.82O alloy (Figure 5D), the Ti-10V-2Fe-3Al-0.11O alloy should possess better damage tolerance. However, its fatigue crack growth thresholds (Δ*K*_0_) could not be estimated because they did not exhibit the normal Paris region at a different stress ratio *R* (Figure 5E). To estimate *K*_0_, the initial Δ*K* of 15.4 and 18.4 MPa/m^1/2^ was used to carry out 10^7^ cycles under a condition of *R* = 0.1. Optical observations found that the crack growth length was less than 0.01 mm. That is, their growth rates (d*a*/d*N*) were less than 10^−12^ m/cycle, which satisfied the typical definition for Δ*K*_0_ that d*a*/d*N* less than 10^−11^ m/cycle. These results reveal that the Δ*K*_0_ of the Ti-10V-2Fe-3Al-0.11O alloy was quite high (≥18.4 MPa/m^1/2^) and its ratio to elastic modulus was one order higher than other metallic materials [24].

### 3.3. The Crack Branch in Tensile Fatigue Tests

Since the Ti-10V-2Fe-3Al-0.11O alloy has an extreme damage tolerance, it is difficult to judge which crack is the main crack (leading to catastrophic failure) and to monitor its evolution with the above bending fatigue test (Figure 5C). Tensile fatigue tests were conducted using single edge-cracked tension specimens. These measurements showed three-level cracks as denoted by blue curves with different widths (Figure 6A). It is clear that the main crack path was similar with the crack in the Ti-10V-2Fe-3Al-0.82O alloy but with more roughness, while the second and third-level cracks propagated away from the nominal growth direction. Digital video records showed that the propagation of the main crack could be blocked by other cracks. Images taken from the video (Figure 6(B1–B4)) demonstrate that the main crack formed in the first 10^6^ cycles (Figure 6(B1)) almost made a stop in the next 10^6^ cycles by propagating other cracks (the blue curves in Figure 6(B2,B3)) and then led to catastrophic failure after an additional 0.2 × 10^6^ cycles (Figure 6(B4)). That is, the crack branch was an effective way to retard the propagation of the main crack.

## 4. Discussion

### 4.1. The Contribution of Elastic Gradient to Damage Tolerance

The plastic deformation zone surrounding the crack tip (i.e., the extent of near-tip plasticity) has a crucial effect on stress gradient, which is the origin inducing the in-situ elastic gradient at the crack tip to suppress crack growth. For conventional metallic materials, the plastic zone is within a few grain diameters in the first stage of crack growth (the near-threshold region) and encompasses many grains in the second stage (the Paris region). Since the stress gradient in the plastic zone is much smaller than its following elastic zone, a large extent of the near-tip plastic zone will have a negative effect on crack resistance.

Three test methods were conducted on the Ti-10V-2Fe-3Al-0.11O alloy to investigate the plastic zone. The first was to reveal whether crack closure mechanisms were operated. The strain gauges were adhered closely to crack fronts to detect the cyclic loading–strain curves. However, these measurements failed to detect the crack closure—for example, a typical group of data (Figure 2F) showed loading and unloading curves that were fully overlapped. The second was unsuccessful in finding a plastic zone by EBSD in high magnification (Figure 5G). These failures drove us to conduct a TEM analysis of the crack tip (Figure 5H). Our observation found a thin layer with nanostructured grains (Figure 5I). SAD analyses, at the points away from the tip, showed the layer having a thickness less than 2 μm (Figure 5J). Additional analyses also found a gradient nano-grained thin layer had formed on both sides of the crack. Since the hot-rolled alloy had a coarser microstructure, it is reasonable to conclude that the microstructure graded layer was formed by highly localized plastic deformation at the micrometer scale during the cycle test. The endurance of such severe plastic deformation at the crack tip gives further support that the elastic gradient contributed greatly to damage tolerance.

### 4.2. The Main Reason for Retardation in Crack Growth

Ti-10V-2Fe-3Al-0.11O is a near-β-type alloy tending to phase transformations at high stress levels from the β phase to the reversible α” martensite and the ω phase [25,26]. Since it has long been recognized that phase transformations at the crack tip lead to a retardation in crack growth, the contribution from the reversible MT was studied using the Ti-10V-2Fe-3Al-0.82O alloy, which has a slightly higher critical stress to the MT than the hot-forged alloy due to the depression effect of grain refinement. The results showed that they exhibited a similar crack path as shown in Figure 5B, but the crack growth threshold (Δ*K*_0_) increased from 13.2 to 15.1 MPa/m^1/2^. Together with the Ti-10V-2Fe-3Al-0.11O alloy having higher critical stress and Δ*K*_0_ over 18.4 MPa/m^1/2^, it is reasonable to conclude that the in-situ elastic gradient is more effective than the MT in retarding crack growth. On the other hand, the further increase of phase stability will deteriorate the nonlinear elasticity as well as the damage tolerance. For example, the oxygen content increased from 0.11% to 0.82%, resulting in a lower Δ*K*_0_ being 11.6 and 5.2 MPa/m^1/2^, particularly in the later alloy exhibiting the normal fatigue crack path. Therefore, a systematic study should be conducted to optimize the balance between nonlinear elasticity and crack growth resistance. Additionally, a new method should also be established to characterize the significant crack deflection and branch (Figure 5 and Figure 6).

### 4.3. The Elastic Gradient Mechanism in Ultrafine Materials

The in-situ elastic gradient mechanism is also valid for ultrafine-graded (UFG) and nanostructured materials. For example, a warm-swaged Ti-10V-2Fe-3Al-0.82O alloy with 0.82O% had duplex microstructures of the UFG β matrix and nano-sized α precipitation (Figure 7A), and exhibited nonlinear elasticity with higher strength (Figure 7B). The three-point bending fatigue test also found that the deflected and branched cracks (Figure 7C) resulted in better damage tolerance (Figure 7D) than other UFG metallic materials showing straight cracks. Due to the precipitation of the nano-sized α phase to strengthen the matrix, the UFG Ti-10V-2Fe-3Al-0.82O alloy had worse damage tolerance than the Ti-10V-2Fe-3Al-0.11O alloy (Figure 7D).

## 5. Conclusions

Our study showed that in-situ induced elastic gradient at the crack tip and defect in materials with elastic deformation behavior exhibited extreme damage tolerance, which can only be realized in natural materials with intricate architectures. This appears to be a perceptive and adaptive mechanism to retard cracks: the higher the stress concentrated at the tip, the larger the elastic gradient to be induced. The retardation is so strong that a gradient nano-grained layer with a thickness of less than 2 μm formed at the crack tip due to the highly localized and accumulated plasticity. Since Ti-10V-2Fe-3Al-0.11O alloy also possesses good biomechanical properties such as a low Young’s modulus of ~55 GPa and a high strength of ~950 MPa, it has great potential for commercial applications to bear heavy loading.

## Figures and Tables

**Figure 1 materials-13-00406-f001:**
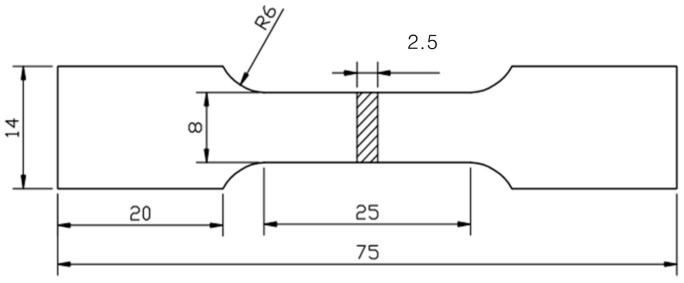
Tensile specimen size at room temperature (unit: mm).

**Figure 2 materials-13-00406-f002:**
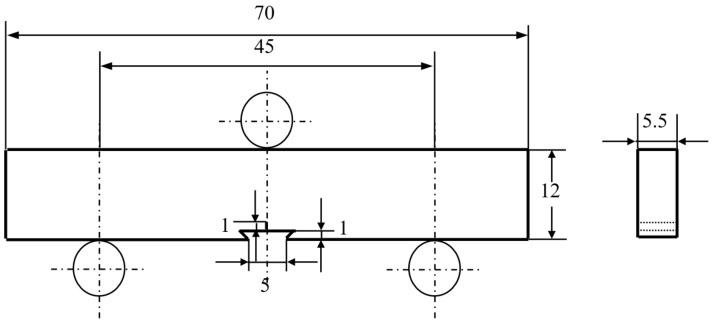
The single edge-cracked bend specimen for the three-point bending fatigue test (unit: mm).

**Figure 3 materials-13-00406-f003:**
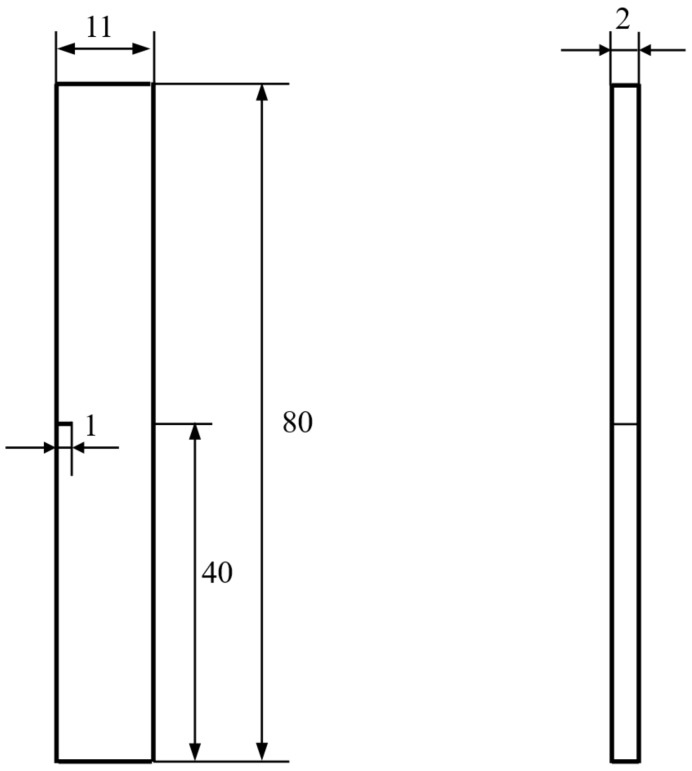
The single edge-cracked tension specimen for fatigue tensile test (unit: mm).

**Figure 4 materials-13-00406-f004:**
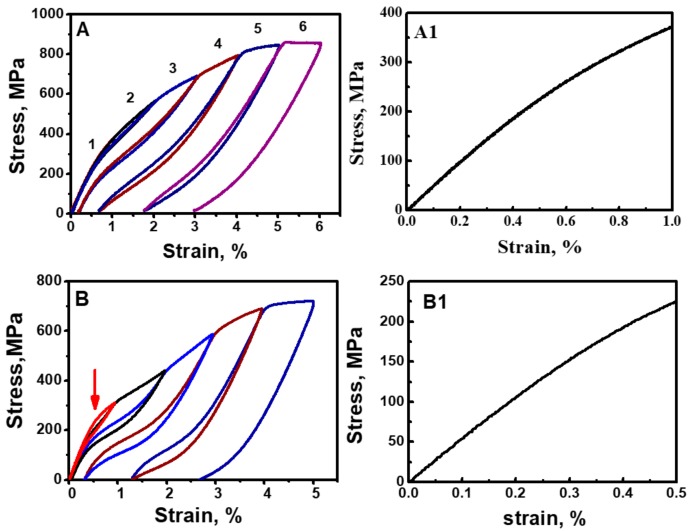
Nonlinear elasticity and elastic gradient. (**A**,**B**) Cyclic tensile loading and unloading stress–strain curves of the Ti-10V-2Fe-3Al-0.11O alloy and the Ti-10V-2Fe-3Al-0.82O alloy, in which the number on the curve represents the cycle number and the arrow shows the critical stress to induce martensite transformation (MT). (**A1**,**B1**) Enlarged stress–strain curves in the low-strain range.

**Figure 5 materials-13-00406-f005:**
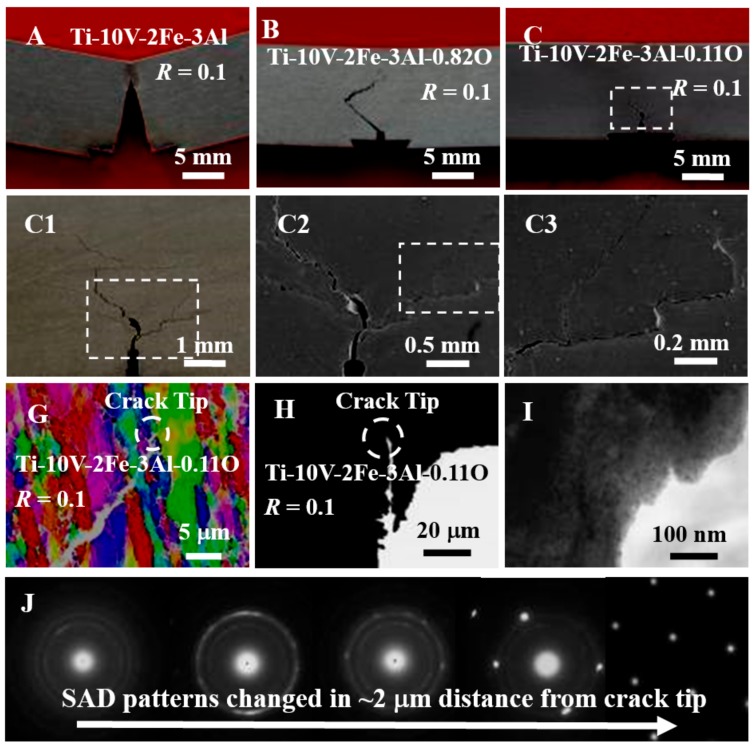
(**A**–**C**) Ti-10V-2Fe-3Al, Ti-10V-2Fe-3Al-0.82, and Ti-10V-2Fe-3Al-0.11 specimens after tests, in which (**C1**–**3**) show the deflected and branched cracks in higher magnifications of (**C**) in turn. (**D**,**E**) Fatigue crack growth rate vs. stress intensity factor range, in which the arrows denote the specimens having d*a*/d*N* less than 10^−12^ m/cycle. (**F**) Loading and unloading–strain curves measured by a strain gauge adhered to the crack front and their variations with cycle numbers. (**G**) Electron backscattering diffraction (EBSD) image of a fatigue crack. (**H**,**I**) TEM images of crack tip in low and high magnifications. (**J**) Variations of selected area diffraction (SAD) patterns away from the crack tip.

**Figure 6 materials-13-00406-f006:**
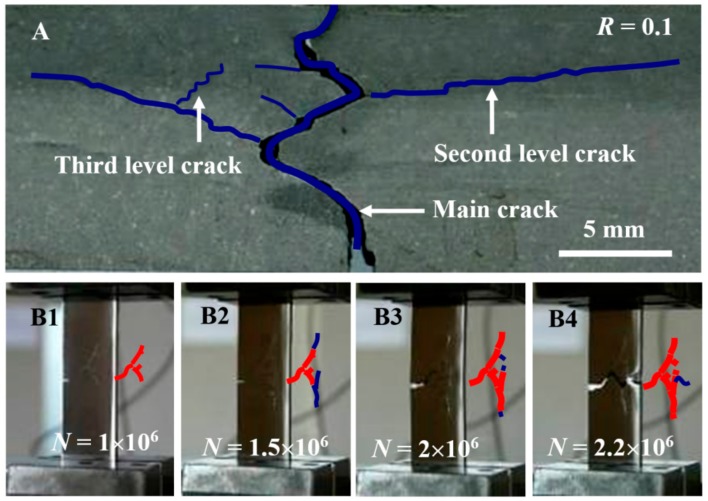
Extreme damage tolerance in tension–tension fatigue test. (**A**) Image of three-level fatigue cracks. (**B1**–**B4**) Crack propagations with cycle numbers *N*, in which the red and blue curves demonstrate the existing cracks in the former image and the extended cracks respectively.

**Figure 7 materials-13-00406-f007:**
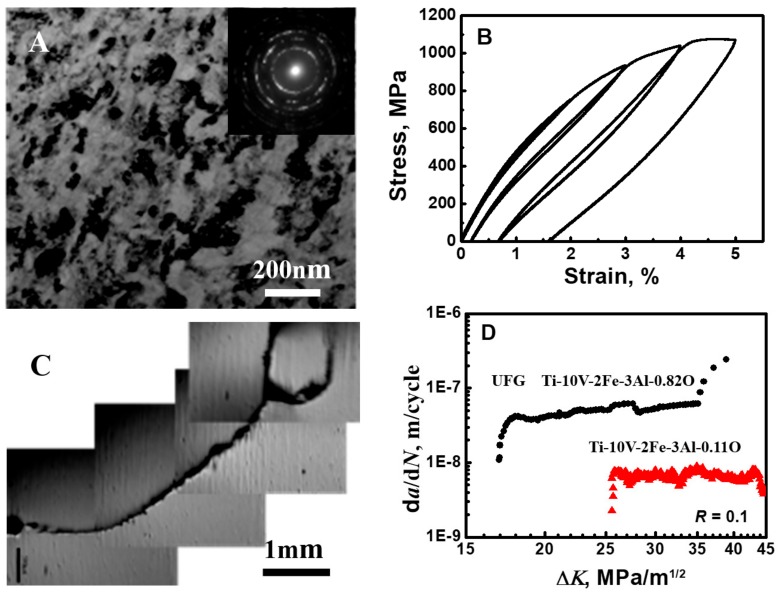
Extreme fatigue damage of ultrafine-grained Ti-10V-2Fe-3Al-0.82O alloy with nano-sized α phase. (**A**) TEM microstructure and the inset SAD pattern. (**B**) The nonlinear elasticity. (**C**) Specimen after three-point bending fatigue test. (**D)** Fatigue crack growth rate vs. stress intensity factor range.

**Table 1 materials-13-00406-t001:** Chemical composition of alloys (wt.%).

Alloy	V	Fe	Al	O	Ti
Ti-10V-2Fe-3Al	9.9	1.80	3.11	0.005	Bal.
Ti-10V-2Fe-3Al-0.11O	10.01	2.05	3.09	0.11	Bal.
Ti-10V-2Fe-3Al-0.82O	10.1	1.98	3.12	0.82	Bal.

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
