# Peer review of "Damage Adaptive Titanium Alloy by In-Situ Elastic Gradual Mechanism"

_materials, 2020, doi:10.3390/ma13020406_

Round 1

Reviewer 1 Report

Dear editor,

Below you can find my comments about the manuscript:

Damage Adaptive Titanium Alloy by In-Situ Elastic Gradual Mechanism

Abstract is to general and needs to be improved Abbreviations are needed, please add immediately after abstract (SEM, TEM, EBSD, etc.) Correct grammar mistakes (free spaces or line → on manuscript line ‘numbers 22, 26, 28, 56, 66, 69, 81, 128, 133, 160, 196, 220, 250, 255, 277, 327) The aim is to general – line 54/page 2. The authors need to explain better how the material were manufactured to ensure the “in-situ” elastic gradient in Ti-alloys – with manufacturing technology or with another approach. It is also the question why they use Ti-alloy, … what was hypothesis??? Part 2 – Materials preparation has to be more precise, it is not well structured, confused,.. The section needs to start with description of chemical composition of initial raw materials, the purity (in %), and other data. Also it is necessary to add T and p of casting process, etc.. Line 62 – the description of equipment for performing of wet chemical and gas analyses is need, also the tolerances of measurements (± %) Table 1 – the chemical composition is not well presented –with which methods the results are obtained, what are tolerances; the number of analysis (parallels) Part 2.2., Figs.1 and 2 – which standard was used for performing of uniaxial cyclic tensile test Part 2.4. – the description of the samples is missing (size, where the sample was taken, etc.), also is missing the purpose of investigation with SEM, TEM, EBSD,…. for what?? Fig.3 – explain in text what are points 1,2,3,4,5,6 on diagram Fig.3A – better presentation is needed with the rest of the text Fig. 4 – text and all presentation is to complicated and confused – please improve Part 4. Discussion – line 255 – add references according to statement Part 4.2 – line 278 – there is no evidence for investigated alloy (Ti-10V-2Fe-3Al-0.11O) in the article, that it has metastable microstructure - if it is possible to add a SEM microstructure with a phase presentation Part 4.2 – please improve – description not reflects the obtained results and the topic presented by the manuscript Part 5 – Conclusion – to general. Please improve with concrete results

Reviewer 2 Report

Dear authors,

In my opinion the study lacks scientific soundness and reproducible results. The number of tests and samples are not mentioned. No images of the samples tested/during testing are provided.

The presentation is quite confusing and editing errors are present throughout the text. It seems like this paper was submitted in a hurry, without checking it or the requirements of the journal.

Please use the format recommended by the journal.

Line 8-9. Please consider using "The aim of this study is to demonstrate" instead of "We demonstrate". Same line 51.

Would you please provide images of the samples (ingots). How many samples were tested? How many tests were perform?

Please give full designation when using abbreviation for the first time: SEM, TEM etc.

Please give more data and manufacturer (name, city, country) for the devices used: MTS landmark testing system, vibration method (JE RT Young s modulus measurement apparatus), YD 28 strain analyzer, LEO SUPRA 35 etc.

Use Figure instead of Fig.

Please provide images during testing. Schematic figures 1 and 2 are not enough to document the study.

Please arrange references according to instructions for authors.

Round 2

Reviewer 1 Report

ok

Reviewer 2 Report

I have no further comments. I will let this pass in the current form, despite the fact that mistakes are still present.